# HBV Infection in HIV-Driven Immune Suppression

**DOI:** 10.3390/v11111077

**Published:** 2019-11-19

**Authors:** Loredana Sarmati, Vincenzo Malagnino

**Affiliations:** Clinical Infectious Diseases, Department of System Medicine, Tor Vergata University, 00133 Rome, Italy; malagninovincenzo@gmail.com

**Keywords:** hepatitis B virus, HIV/HBV coinfection, end-stage liver disease, hepatocellular carcinoma, occult HBV infection

## Abstract

Worldwide, approximately 10% of all human immunodeficiency virus (HIV)-infected people are also chronically coinfected with hepatitis B virus (HBV). HBV infection has a poor prognosis in HIV-positive people and has been documented by an increased risk of developing chronic HBV infection (CHB), progression to liver fibrosis and end-stage liver disease (ESLD) and evolution of hepatocellular carcinoma (HCC). Furthermore, in HIV patients, HBV-resolved infection is often associated with the appearance of HBV-DNA, which configures occult HBV infection (OBI) as a condition to be explored in coinfected patients. In this narrative review we summarize the main aspects of HBV infection in HIV-positive patients, emphasizing the importance of carefully considering the coinfected patient in the context of therapeutic strategies of antiretroviral therapy.

## 1. Introduction

Approximately 360 million individuals worldwide are chronically infected with hepatitis B virus (HBV); among these, approximately 1 million die because of end-stage liver disease (ESLD) or hepatocellular carcinoma (HCC). In 2018, it was reported that 37.9 million people globally were living with HIV [1], and it is believed that approximately 10% of all HIV-infected people are also chronically coinfected with HBV, with differences related to the variability in prevalence of HBV in the areas described in a World Health Organization (WHO) classification [2].

HIV/HBV coinfection status determines a constant interaction between the two viruses that modifies the natural history of both monoinfections. In particular, HBV infection in HIV-positive patients is more likely to become chronic, despite a greater risk, after infection, of incurring acute hepatitis. On the other hand, in HIV-positive patients, the presence of HBV infection is associated with a lower number and a lower recovery of CD4 lymphocytes as well as lower virologic response in a course of combined antiretroviral therapy (cART) [3].

A number of studies have shown that HBV infection adversely affects the course of liver disease and HIV infection in coinfected subjects [4]; therefore, a number of prognostic and therapeutic considerations regarding HBV coinfection in HIV-positives patient have been raised. 

This narrative review explores the most recent issues from the literature on HBV/HIV coinfection, focusing on the most relevant clinical and laboratory aspects concerning the outcome of HBV infection in HIV-infected patients.

## 2. Methodology

The purpose of this article is to provide a synthesis of the current relevant aspects of HIV/HBV coinfection; it was conducted with a narrative review methodology, highlighting the main results. The literature search was based on a search strategy for English-language literature in PubMed and the Cochrane Library. The research was performed by the authors, who independently reviewed all articles for study inclusion. The search terms used either alone or in combination were ‘hepatitis B virus,’ ‘HBV,’ ‘human immunodeficiency virus’, ‘HIV infection, ‘HBV and HIV coinfection’, ‘occult HBV infection’ (OBI), ‘antiretroviral therapy’, ‘antiviral therapy’, ‘liver/hepatitis’, ‘liver damage’, ‘cirrhosis’ and ‘hepatocellular carcinoma’.

## 3. Main Aspects of HBV Infection in the HIV-Infected Patient

Active HBV replication is known to be a risk factor for liver-related morbidity and mortality in HBV monoinfected patients [4]. In patients with HIV/HBV, the coinfection accelerates HBV-related liver damage, with faster liver fibrosis progression and more frequent development of ESLD and HCC compared to HBV monoinfected patients [5,6,7]. Conversely, the presence of a simultaneously active HBV infection affects the viro-immunological status of HIV/HBV-coinfected patients, which is usually characterized by a lower CD4-positive lymphocyte count at the onset and a slower CD4 cell count recovery after the start of cART [8,9]. In addition, lower CD4 lymphocyte counts have been associated with progression of liver fibrosis during HIV/HBV coinfected patients [2,3]. Despite the introduction of cART, with the consequent benefit of HIV viremia suppression and immunological recovery, the differences in the outcome between HIV monoinfected and HIV/HBV coinfected populations still persist. Moreover, recent studies have shown that even the condition of resolved HBV infection (presence of anti Hepatitis B c antigen [anti-HBc] in the absence of Hepatitis B surface antigen [HBsAg]) can often be associated with the appearance of detectable HBV-DNA in HIV-positive patients (occult HBV infection (OBI)). In HIV-positive subjects, the OBI condition has been associated with a low number of CD4-positive lymphocytes, elevation of alanine aminotransferase and more frequent AIDS-defining illnesses [9,10].

In the sections below, the different aspects of HIV/HBV coinfection are analyzed (Table 1).

## 4. HBV Replication and Immune Control in HIV/HBV Coinfection

In the natural course of HBV mono-infection, the formation of the episomal HBV covalently closed circular DNA (cccDNA) and the integration of the HBV DNA supported the production of HBsAg and HBV replication. HBsAg has an inhibitory impact on the adaptive immune response and on the production of anti-HBs antibodies, which allows definitive protection and recovery from HBV infection.

Little is known about the outcome of the natural course of HBV infection in coinfected people. Accelerated liver disease progression and HCC evolution, which has been documented in HIV/HBV infected patients, is likely to be related to high HBV-DNA levels [27] and poor immunological control of HBV replication associated with the low CD4 positive cell counts, which are typical of HIV/HBV coinfected subjects. A greater tendency to synchronize HBV infection with poor HBeAg seroconversion has been demonstrated in coinfected Nigerian people before the onset of cART [12]. In this study, higher HBV DNA and detectable HBeAg levels were independently associated with lower CD4+ T cell counts. 

The antiviral treatment results in a slowing down of the production cycle of HBsAg and is able to counteract the HBsAg-driven inhibition of the immune response, but not to completely interrupt the HBV replicative process. Surprisingly, many studies have observed that in HIV-positive patients with chronic HBV hepatitis (CHB), the proportion of patients experiencing HBsAg loss or conversion has been greater than in monoinfected patients [13,14,15,16,17,18,28]. In particular, a greater tendency of HBsAg loss has been described along with, interestingly, a lower number of CD4 cell counts pre-cART and a greater increase in CD4 cells over the course of effective cART in patients observed for a long duration of follow-up [15,16,17,18,28]. The more frequent HBsAg loss and anti-HBs conversion observed in patients with HIV-HBV coinfection have been explained by the immuno-substitutive thrust given by cART, which promotes an increase in the proportion of CD4-positive cells, allowing a sudden recovery of adaptive immunity and subsequently accelerating the production of protective antibodies. Many authors have described HIV-positive patients who seroconverted for HBV after hepatic flares, which are related to HBsAg seroclearance and are an aspect of HBV-related immune reconstitution inflammatory syndrome (IRIS) [29]. Occasional cases of fatal acute liver failure that occur in the setting of HIV/HBV coinfection after cART initiation have also been reported [30].

Regarding the loss of HBsAg, many papers have described an association between pre-cART HBsAg levels and the HBsAg loss frequency during follow-up [15,18,28,31,32,33]. In particular, in a prospective cohort study from two randomized-control trials in Côte d’Ivoire, HBsAg levels ≤ 100 or ≤ 10 IU/mL at 12 months post-cART initiation better correlated with HBsAg seroclearance. Furthermore, low levels of HBsAg, although not clearly associated with HBsAg clearance, are predictive of HBeAg seroconversion [15,18]. Also lower baseline HBV-DNA was found associated with increased HBsAg seroclearance (functional cure) [13].

## 5. OBI and HIV Infection

In the context of HIV-HBV coinfection, it is important to highlight the framework of OBI. OBI is defined as the presence of replication-competent HBV DNA (i.e., covalently closed circular DNA [cccDNA]) in the liver and/or HBV DNA in the blood of people who test negative for HBsAg [19]. This is a condition in which, in the absence of HBsAg, viral replication at the intrahepatocyte level persists, with the consequence of a high formation of cccDNA and increased integration of HBV DNA [19,34,35]. The prevalence of OBI has been reported from 1% of HIV-positive individuals in the United States to >15% of HIV-positive individuals in countries such as South Africa [36]. OBI has already been associated with increased progression of hepatic fibrosis, ESLD and HCC, and it has been followed by possible HBV reactivation under severe immunosuppression [11,35,36,37]. However, there are gaps in the literature regarding the real incidence and outcome of OBI in the HIV population. In particular, the condition of “isolated anti-HBc” is a commonly observed condition in HIV-positive patients, probably due to the spontaneous loss of anti-HBs in patients with resolved HBV infection, or as a result of the delayed appearance of anti-HBs in subjects losing HBsAg under tenofovir (TDF) treatment. A number of studies have demonstrated that anti-HBc HIV-positive subjects showed the occasional presence of detectable levels of HBV-DNA in plasma [11,38,39] either in the course of cART following an interruption of antiretroviral therapy, or in relation to the appearance of transient viremia and, therefore, the appearance of true OBI. Conversely, a limited control of HIV infection treatment was demonstrated by recent papers in coinfected anti-HBc positive patients [40,41,42].

A careful follow-up in HIV/HBV coinfected patients with a systematic periodic study of complete HBV serology, a hepatic ultrasound, a non-invasive fibrosis evaluation (FibroScan) and/or the invasive (liver biopsy) study of hepatic tissue, has been indicated by some international guidelines [20,21]. However, there is a lack of studies describing and suggesting specific laboratory or instrumental assessments to better evaluate HBV infection progression and liver disease evolution in HIV-positive patients with signs of a resolved HBV infection.

## 6. Liver Damage during HIV/HBV Coinfection.

In monoinfected CHB patients, the progression of liver fibrosis and hepatic injury is generally improved with the use of analogous nucleosides (NAs), particularly drugs such as lamivudine (LAM), emtricitabine (FTC), tenofovir dipivoxil fumarate (TDF) and tenofovir alafenamide (TAF) [22]. In contrast, in the coinfected HIV/CHB patients, studies performed through biochemical and histological control [23] or through transient elastography [22,23,24] have shown only a short-term benefit of NA use in stopping the progression of liver fibrosis. A large, recent US retrospective study analyzed the predisposing factors for the evolution of liver injury and the occurrence of ESLD in 3573 HIV/CHB patients, and demonstrated that a low CD4+ count (CD4+ < 200/mmc aHR = 1.58 [CI 95% 1.36–4.91] and CD4+ 201–499/mmc, aHR 1.75 [CI 95% 1.04–2.39] vs. CD4+ > 500/mmc) and higher FIB-4 at the start of follow-up (>3.25: aHR = 9.79 [CI 95% 5.73–16.74]; 1.45–3.25: aHR = 3.20 [CI 95% 1.87–5.47] vs. FIB-4 < 1.45), in addition to diabetes and non-black/non-Hispanic race, were factors associated with an increased risk of liver disease progression. In this study, HIV viremia did not seem to be associated with liver complications (aHR = 0.56 [CI 95% 0.35–0.91]) [25]. Moreover, Boyd et al. [43] evaluated the evolution of liver fibrosis in a nine-year prospective study using a METAVIR score during treatment of 167 HIV/HBV coinfected patients with TDF. Only 15% of the study patients with F3-F4 baseline fibrosis achieved regression of fibrosis at the end of follow-up, and the general benefit was found only in the first years of TDF intake. The absence of regression of hepatic fibrosis was subsequently confirmed in the same cohort, while demonstrating a low incidence of morbidity and mortality related to hepatic events (IR = 1.05/100 person-years) [44]. Nonetheless, the French group confirmed that a high value of CD4+ cells at baseline was more frequent in patients undergoing regression of liver fibrosis, and highlighted that the duration of cART (in particular, treatments containing protease inhibitors [PIs]), was significantly associated with progression to a severe stage of F3-F4 fibrosis in patients with a METAVIR score at baseline of F0, F1 or F2 (OR 1.07, CI 95% 1.00–1.15 and OR 2.41, CI 95% 1.38–4.19, respectively) [43]. The role of cART is to be interpreted as the accumulation of liver toxicity generated by the possible administration of obsolete drugs such as zidovudine and ritonavir-boosted saquinavir and didanosine; all of these drug agents are known to increase the risk of lipodystrophy, steatosis, steatohepatitis, insulin resistance and the progression of liver fibrosis [45].

Ultimately, despite effective antiviral treatment on both viruses, coinfected HIV/HBV patients have still experienced an increased progression of liver fibrosis during follow-up and HIV viral suppression [45]. 

A recent study on African subjects with HIV and CHB demonstrated a high risk of mortality, despite an early cART initiation, in the presence of HBV-DNA levels > 2000 UI/mL [46].

Nonetheless, the data available show that a prompt diagnosis of coinfection together with a rapid onset of cART are indispensable for slowing liver damage, fibrosis and evolution towards ESLD in the context of a growing CD4+ count and rapid and constant viral suppression of HIV and HBV.

## 7. HCC in HIV/HBV Coinfected Patients

Liver disease, particularly HCC, remains a major cause of hospitalization for HIV-infected patients. Certainly, the phenomenon is linked to the high prevalence of hepatotropic viral infection in this population and, in a portion of this population, to an increase in alcohol intake and drug addiction [25,46,47]. Regarding the onset of HCC in HIV/HBV-coinfected patients, a recent retrospective study based on data from four major European cohorts calculated an incidence of HCC of 5.9 per 1000 patient-years PY (95% CI 3.60–9.10) in HIV/HBV cirrhotic patients, which is significantly higher than that of HIV/HBV non-cirrhotic patients (1.17 per 1000 PY (CI 95% 0.56–2.14)) [26]. Moreover, patients over 45 years of age at the start of TDF treatment had an increased risk of developing HCC compared to younger patients. Liver cirrhosis remains the most important risk factor for the evolution of HBV infection towards HCC, but, as shown in the study above, HCC occasionally appears during HBV infection even in the absence of liver cirrhosis. The mechanism underlying this phenomenon is not known, although it may be related to the persistence of HBV-integrated intracellular DNA. In the aforementioned OBI condition, the risk of HCC occurrence has only been studied in HBV-monoinfected patients [48,49], and there is a lack of data from HIV/HBV coinfected patients. Factors that have been related to an increased risk of HCC evolution in patients with HIV/CHB coinfection are the pre-therapy level of HBV DNA and the titer and rate of quantitative HBsAg decline from the time of the therapy’s initiation [5,49,50]. It is important to note that the incidence of HCC has remained stable over time among coinfected individuals on TDF treatment, whereas it has increased steadily among those not on TDF therapy. However, the condition of liver cirrhosis has also had a great impact; in fact, among patients on TDF, the incidence of HCC is 5.9 per 1000 PY (95% CI 3.60–9.10) in cirrhotics and 1.17 per 1000 PY (0.56–2.14) among non-cirrhotics [26].

## 8. Drug-Resistant and Immune-Escape HBV Mutants in HIV/HBV Coinfected Patients

In HIV/HBV coinfected patients, antiviral therapy reduces the progression to liver fibrosis and, consequently, the development of ESLD and HCC. Under drug pressure, the occurrence of drug-resistant HBV strains is, however, possible. Moreover, a poor anti-HBV drug susceptibility or a real drug resistance, may be naturally present in some HBV genotypes as a consequence of its genetic variability. Genotypic variety of HBV is a result of gradual evolution in the absence of drug selective pressure, and is linked to the natural history and geographical origins of the viral strains. In this context, very few studies have analyzed the impact of HBV genotypes on the evolution of hepatic disease in HIV/HBV coinfection [43,51]. Typically, HBV genotype E constitutively harbors the modification of HBsAg that increases its potential to evade host immune response, and it shows a natural resistance to many anti-HBV treatments. The genotype G of HBV has also been correlated worse outcomes and greater risk of evolution in liver fibrosis in coinfected patients.

Due to the virus’s attempt to escape the host’s immune response (Precore, Core, Basal Core Promoter mutations), the appearance of HBV phenotypic variability is induced by low genetic barrier treatments (Pol mutants) or by anti-HBV immunoglobulin (immune-escape mutants). Regarding the emergency of drug resistant viral strains, some recent publications have reported their relevance in HIV/HBV coinfection. LAM, commonly contained in cART regimens in developing countries [52,53], has been shown to induce resistance mutations after a few years of use [54,55]. In a study on HIV/HBV coinfection, Matthews and colleagues [56] showed a development of LAM resistance in 50% of patients treated with LAM for less than 24 months and in 94% of those treated for more than 48 months The authors reported the emergence of single, double or triple mutations, also in combination (rtM204V/I, rtM204V/ + rtL180M, rtV173L + rtL180M + rtM204V). Because of the overlapping of genes, the triple mutation rtV173L + rtL180M + rtM204V leads to the emergence of surface mutations sE164D/sI195M that could result in strains with characteristics of a vaccine escape, which could be potentially transmissible to vaccinated patients. Moreover, prolonged treatment with LAM-containing cART of HIV/HBV coinfected patients harboring LAM-resistant HBV causes an accumulation of drug mutations that can confer resistance to Entecavir. 

Only a few case reports have described the potential emergence of genotypic mutations in patients treated with TDF [57,58,59], although many longitudinal studies have well described the substantial absence of resistance selection in patients treated with TDF long-term [25,60,61].

## 9. Conclusions

HBV infection in HIV-positive patients is frequently associated with a worsened immuno-virological setting. HBV infection itself has unique characteristics in HIV-infected patients, such as a lower tendency to resolve the infectious state and a greater tendency to evolve in a chronic phase, as well as an increased progression to hepatic fibrosis. Patients with HIV/HBV coinfection are at greater risk of evolution in ESLD and HCC compared to HIV- or HBV-monoinfected patients. Therefore, the maintenance of anti-HBV drugs active within the cART composition are mandatory in order to contain these risks. 

Little is known about the condition of resolved HBV infection and potential OBI in the context of HIV infection, and future in-depth studies are needed to better understand if occasional HBV viremia may result in hepatic liver injury in this type of patient. Moreover, with the current trend of using two drugs in the cARTcomposition and/or NUC-sparing cART regimes, it could be useful to understand whether the choice of these therapeutic strategies requires caution in HIV-positive patients with a previous resolved HBV infection.

## Figures and Tables

**Table 1 viruses-11-01077-t001:** Main characteristics of patients with HBV/HIV coinfection.

Characteristics	Studies’ Results and Prognosis of HIV/HBV Coinfection	References
**Low CD4 cell count**	CD4 nadir < 100 cell/mmc, associated with death and impaired CD4 recovery during the cART *	[5,7,11]
**Higher risk of AIDS and death**	CHB ° significantly increased liver-related mortality in HIV patients	[3,5]
**Increased HBV ^§^ replication**	High HBV DNA levels in half of HIV ^^^/CHB patients	[6,10]
**Increased likelihood of loss of anti-HBs ^&^**	Functional cure more frequent in HIV/HBV patients and correlated with low pre-treatment HBsAg levels	[12,13,14,15,16,17,18]
**Increase OBI °° condition**	OBI associated with low CD4 cells, high ALT elevation and more frequent AIDS illness	[8,9,11,19]
**Increased liver disease progression**	A scarce benefit of anti-HBV treatment on liver fibrosis. Low CD4 count and diabetes associated factors	[20,21,22,23,24,25]
**Increased risk for HCC ****	HCC incidence 5.9 per 1000 person years	[26]

* cART, combined antiretroviral therapy; ° CHB, chronic HBV infection; ^§^ HBV, hepatitis B virus; ^^^ HIV, human immunodeficiency virus; ^&^ anti-HBs, anti antigen S of HBV antibodies, °° OBI, occult HBV infection; ** HCC, hepatocellular carcinoma.

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
