# Peer review of "HBV Infection in HIV-Driven Immune Suppression"

_viruses, 2019, doi:10.3390/v11111077_

Round 1

Reviewer 1 Report

The review by Sarmati L. and Malagnino V. has been submitted in the frame of the Special issue “Hepatitis B virus reactivation” that is aimed at providing new and updated insights into virological and immunological mechanisms underlying HBV persistence and factors promoting viral reactivation.

The paper explores the existing literature on HBV infection in HIV-driven immune suppression. However, it does not contain very substantial and critical new information in the area of reactivation of HBV infection in the HIV-infected patient that should be the central argument of the special issue. In particular, a very important aspect in this field that is lacking in this review is the emergence of drug resistance mutations in the HBV genome in ART-experienced HIV-coinfected patients that has an impact on the therapeutic schedule to administer to the HIV/HBV infected patient to minimise the emergence and spread of antiviral drug resistance. In addition, the emergence of HBV immune escape mutants should be mentioned and discussed.

Thus, I would suggest to include in the review also an updated information on the drugs to use in the treatment of these patients and on the possibility of emergence of drug resistance and vaccine escape mutants.

Finally, English should be improved, in particular in the Abstract section where there are several grammatical and syntactical errors.

Author Response

Manuscript ID: viruses-635085

Type of manuscript: Review

Title: HBV infection in HIV-driven immune suppression

Reviewer 1

The review by Sarmati L. and Malagnino V. has been submitted in the frame of the Special issue “Hepatitis B virus reactivation” that is aimed at providing new and updated insights into virological and immunological mechanisms underlying HBV persistence and factors promoting viral reactivation.

The paper explores the existing literature on HBV infection in HIV-driven immune suppression. However, it does not contain very substantial and critical new information in the area of reactivation of HBV infection in the HIV-infected patient that should be the central argument of the special issue. In particular, a very important aspect in this field that is lacking in this review is the emergence of drug resistance mutations in the HBV genome in ART-experienced HIV-coinfected patients that has an impact on the therapeutic schedule to administer to the HIV/HBV infected patient to minimise the emergence and spread of antiviral drug resistance. In addition, the emergence of HBV immune escape mutants should be mentioned and discussed.

Thus, I would suggest to include in the review also an updated information on the drugs to use in the treatment of these patients and on the possibility of emergence of drug resistance and vaccine escape mutants.

Answer- Regarding the reactivation of HBV in coinfected patients, most of the studies published on this topic concern patients with resolved HBV (anti-HBc) or OBI, this topic is mentioned in the chapter OBI and HIV infection.

In accordance with the suggestions of the reviewer, a paragraph on HBV viral strains with resistance to antiviral drugs and with characteristics of immune escape is now present in the (lines 193-228). Related articles have been added to the references list.

Reviewer 2 Report

The authors present a review on HIV/HBV coinfection, focused on the HBV part, and its evolution in the coinfection setting, more specifically on the clinical and laboratory aspects. It is rather complete, well presented, with complete and recent references. One could just regret a too short HBV therapeutic part.

Introduction:

it is stated that 10% of HIV-infected subjects are co-infected with HBV; this is a worldwide estimation, with geographical differences, and a rather lower (7-8%) prevalence in Europe or North-America; to be detailed. HIV/HBV coinfection could raise questions of interactions between the 2 viruses in both directions, which could be more clearly specified.

Main aspects of HBV infection in the HIV-infected patient:

a slower CD4 cells recovery is effectively observed in HIV/HBV coinfected patients, but is “normal” again when HBV is controlled by the antiviral treatment (see reference 8 by Wandeler et al), i.e. similar to HBV uninfected subjects. Are there any data on this slower immune recovery, according to the liver status (higher impact of an advanced fibrosis) ? Table 1 is not very easy to read …

HBV replication …

the part on HBsAg loss and clearance is very interesting; it is stated that low levels of this Ag after 12 months of ARV treatment were correlated with clearance, it should be added that lower baseline values were also predictive, which is an important information.

OBI and HIV infection

lines 113-115: I don’t in the references 32 and 37 any clear guideline recommending a careful follow-up of HBV, and what kind of follow-up, specifically in OBI. There are no HBV treatment comment in this section (and reference 32 is evaluating the impact of HBV treatment).

HCC in HIV/HBV coinfected patients

based on reference 49, I would point out the positive effect of TDF treatment on the risk of HCC development.

Author Response

Manuscript ID: viruses-635085

Type of manuscript: Review

Title: HBV infection in HIV-driven immune suppression

Reviewer 2

The authors present a review on HIV/HBV coinfection, focused on the HBV part, and its evolution in the coinfection setting, more specifically on the clinical and laboratory aspects. It is rather complete, well presented, with complete and recent references. One could just regret a too short HBV therapeutic part.

Introduction:

it is stated that 10% of HIV-infected subjects are co-infected with HBV; this is a worldwide estimation, with geographical differences, and a rather lower (7-8%) prevalence in Europe or North-America; to be detailed.

Answer : We agree with the reviewer and, at the end of the first paragraph (lines 28-29),  a sentence and a bibliographic reference on the different world prevalence of coinfection have been added.

HIV/HBV coinfection could raise questions of interactions between the 2 viruses in both directions, which could be more clearly specified.

Answer : according to reviewer’s request, a sentence which explains the bidirectional interactions between the 2 viruses was added to the text (lines 33-36)

Main aspects of HBV infection in the HIV-infected patient:

a slower CD4 cells recovery is effectively observed in HIV/HBV coinfected patients, but is “normal” again when HBV is controlled by the antiviral treatment (see reference 8 by Wandeler et al), i.e. similar to HBV uninfected subjects. Are there any data on this slower immune recovery, according to the liver status (higher impact of an advanced fibrosis) ?

Answer : according to reviewer’s request, at lines 60-61, it is now present a sentence and a reference on the impact of low CD4 recovery and the development of  liver fibrosis

4.Table 1 is not very easy to read …

 Answer - the table has been and improved

HBV replication …

the part on HBsAg loss and clearance is very interesting; it is stated that low levels of this Ag after 12 months of ARV treatment were correlated with clearance, it should be added that lower baseline values were also predictive, which is an important information.

Answer – at lines 107-109 a phrase on low level of HBsAg and HBeAg seroclearance has been added

OBI and HIV infection

lines 113-115: I don’t in the references 32 and 37 any clear guideline recommending a careful follow-up of HBV, and what kind of follow-up, specifically in OBI. There are no HBV treatment comment in this section (and reference 32 is evaluating the impact of HBV treatment).

Answer – In the chapter on OBI condition and HIV, it is highlighted the current lack of strong data to support any future indications concerning the monitoring of patients with HIV and possible OBI. Currently there are no data that support guideline recommendations on this field, which instead are present for chronic HBV infection in HIV.  The two references 33 and 38 concern chronic HBV infection in HIV a structured follow up has been established by guidelines.

7.HCC in HIV/HBV coinfected patients

based on reference 49, I would point out the positive effect of TDF treatment on the risk of HCC development.

Answer – in accordance with the reviewer' request, at lines 192-197,  a sentence that reports the positive effects of TDF on the development of HCC is now present.

Round 2

Reviewer 1 Report

The authors have addressed all the issues I raised in my previous revision. As required, a chapter on drug-resistant and immune-escape mutants is now present in the manuscript.

The English language has been revised.

Please, find following some proposed changes that can help to improve the clarity of the text.

Line 13: change “was” with “is”.

Line 33: change “was” with “is”.

Line 40: change “explored” with “explores”.

Line 40: change “literature” with “the literature”

Lines 88-90: suggested sentence: “The antiviral treatment results in a slowing down of the production cycle of HBSAg and is able to counteract the HBSAg-driven inhibition of the immune response, but not to completely interrupt the HBV replicative process”.

Line 92: delete “that”.

Line 106: change “correlate” with “correlated”.

Line 110: delete “were”.

Lines 184-184: suggested sentence: “Patients over 45 years of age at the start of TDF treatment had an increased risk of developing HCC compared to younger patients”.

Line 189: delete the comma after “while”.

Line 204: change “drugs” with “drug”.

Lines 219-222: suggested sentence: “In a study on HIV/HBV coinfection, Matthews and colleagues showed the development of LAM resistance in 50% of patients treated with LAM for less than 24 months and in 94% of those treated for more than 48 months.

Author Response

Manuscript ID: viruses-635085

Type of manuscript: Review

Title: HBV infection in HIV-driven immune suppression

Reviewer 1

Comments and Suggestions for Authors

The authors have addressed all the issues I raised in my previous revision. As required, a chapter on drug-resistant and immune-escape mutants is now present in the manuscript.

The English language has been revised.

Please, find following some proposed changes that can help to improve the clarity of the text.

Line 13: change “was” with “is”.

Answer: the modification requested is now present in the text

Line 33: change “was” with “is”.

Answer: the modification requested is now present in the text

Line 40: change “explored” with “explores”.

Answer: the modification requested is now present in the text

Line 40: change “literature” with “the literature”

Answer: the modification requested is now present in the text

Lines 88-90: suggested sentence: “The antiviral treatment results in a slowing down of the production cycle of HBSAg and is able to counteract the HBSAg-driven inhibition of the immune response, but not to completely interrupt the HBV replicative process”.

Answer: the phrase has been changed as suggested by the review

Line 92: delete “that”.

Answer: the modification requested is now present in the text

Line 106: change “correlate” with “correlated”.

Answer: the modification requested is now present in the text at line 109

Line 110: delete “were”.

Answer: the modification requested is now present in the text

Lines 184-184: suggested sentence: “Patients over 45 years of age at the start of TDF treatment had an increased risk of developing HCC compared to younger patients”.

Answer: the phrase has been changed as suggested by the review

Line 189: delete the comma after “while”.

Answer: the modification requested is now present in the text

Line 204: change “drugs” with “drug”.

Answer: the modification requested is now present in the text

Lines 219-222: suggested sentence: “In a study on HIV/HBV coinfection, Matthews and colleagues showed the development of LAM resistance in 50% of patients treated with LAM for less than 24 months and in 94% of those treated for more than 48 months.

Answer- the phrase has been changed as suggested by the review

.
